# Growth, Gas Exchange, and Phytochemical Quality of Nasturtium (*Tropaeolum majus* L.) Subjected to Proline Concentrations and Salinity

**DOI:** 10.3390/plants14030301

**Published:** 2025-01-21

**Authors:** Vitor Araujo Targino, Thiago Jardelino Dias, Valéria Fernandes de Oliveira Sousa, Mariana de Melo Silva, Adjair José da Silva, João Everthon da Silva Ribeiro, Ramon Freire da Silva, Diego Silva Batista, Juliane Maciel Henschel, Mailson Monteiro do Rêgo

**Affiliations:** 1Graduate Program in Agronomy, Federal University of Paraíba, Areia 58397-000, PB, Brazil; mariana.melo@academico.ufpb.br (M.d.M.S.); adjair.engagronomo@gmail.com (A.J.d.S.); ramonsilvagro@gmail.com (R.F.d.S.); diegoesperanca@gmail.com (D.S.B.); julianemhenschel@gmail.com (J.M.H.); mailson@cca.ufpb.br (M.M.d.R.); 2Academic Unit of Agrarian Sciences, Federal University of Campina Grande, Pombal 58840-000, PB, Brazil; 3Graduate Program in Phytotechnics, Federal Rural University of Semi-Arid, Mossoró 59625-900, RN, Brazil; j.everthon@hotmail.com

**Keywords:** abiotic stress, edible flowers, nasturtium, osmoprotectants, salt stress

## Abstract

Salinity is a significant challenge for agriculture in semi-arid regions, affecting the growth and productivity of plants like *Tropaeolum majus* (nasturtium), which is valued for its ornamental, medicinal, and food uses. Salt stress disrupts the plant’s biochemical, physiological, and anatomical processes, limiting its development. This study investigates the potential of proline as an osmoprotectant to mitigate the effects of salt stress on nasturtium’s growth and physiology. A completely randomized factorial design was employed, testing five levels of electrical conductivity (0.0, 1.50, 3.00, 4.5, 6.5 dS m^−1^) and four proline concentrations (0.0, 5.00, 10.0, 15.0 mM) with six replicates. The results showed that proline application, particularly at 15.0 mM, enhanced growth parameters such as leaf number, stem diameter, and root length. At moderate salinity (3.0 dS m^−1^), proline significantly improved gas exchange, increasing net photosynthesis, transpiration, and stomatal conductance. Additionally, proline reduced the negative impact of salt stress on the fresh mass of leaves, stems, and roots, and increased both the mass and number of flowers. Proline also elevated the levels of total phenolic compounds and vitamin C while reducing soluble sugars, particularly under moderate salt stress (4.75 dS m^−1^). Overall, applying 15.0 mM proline shows promise for enhancing the biomass accumulation, flower production, and overall quality of nasturtium under saline conditions.

## 1. Introduction

*Tropaeolum majus* L. (Tropaeolaceae), also known as ‘capuchinha’ and ‘chaguinha’ in Brazil, is a versatile plant with edible, ornamental, and medicinal properties [1]. Its flowers, which can be single or double, grow up to 2–3 cm, while the plant reaches about 30–40 cm [2]. The vibrant nasturtium flowers in shades of red, orange, and yellow have a distinctive spicy flavor and are ideal for enhancing salads, sauces, grilled dishes, and other culinary preparations [3]. In addition, nasturtium flowers are rich in bioactive compounds, including flavonoids such as quercetin and isoquercitrin, fatty acids such as oleic and linoleic, vitamin C, and benzyl thiocyanate [4].

Nasturtium leaves and flowers have been used as a therapeutic resource for several conditions, including hypertension, inflammation, urinary tract infections, wounds, gallbladder disorders, aphrodisiac properties, and in the treatment of chronic diseases, such as obstructive pulmonary disease, kidney and bladder infections, in addition to being considered to have anticarcinogenic potential [5]. Both flowers and leaves are recognized as a valuable dietary source of lutein, contributing to the reduction in the risk of macular degeneration. In addition, they have diuretic and antihypertensive properties, with beneficial effects on diabetes control [6]. However, nasturtium cultivation can be significantly impacted by adverse environmental conditions, such as salinity.

Salinity compromises several metabolic processes in plants, negatively affecting gas exchange, nutritional absorption and balance, and osmotic balance, causing decreased growth [7]. These effects occur due to the low water availability caused by reduced osmotic potential, changes in enzyme activity, and oxidative stress caused by the increased production of reactive oxygen species (ROS) and ionic toxicity, which is mainly caused by Na^+^ and Cl^−^ [8,9]. Nasturtium is a plant species that is extremely sensitive to salinity, as shown in the studies presented in [10,11,12]. According to [10,12], salt stress (0 to 80 mM NaCl) caused damage to the photosynthetic apparatus, which led to a decrease in gas exchange and, consequently, in the biomass of nasturtium plants. Also, according to [10], the increase in salt stress reduced the contents of total phenolic compounds, reducing sugars, and non-reducing sugars, as well as the number of flowers and plant growth.

In view of these challenges, the demand for products that can reduce and attenuate the effects of salt stress on plants is growing. Among them, the use of proline stands out. Proline plays an important role as an osmolyte, acting as a chelator for metals and activating the cellular antioxidant system to cope with stressful conditions such as water and nutrient scarcity [13]. In addition, proline has a protective effect against phospholipids, plasmalemma, mitochondria, and plastid membranes [14]. The exogenous application of proline can help plants develop and improve their photosynthetic activity and mineral nutrition by regulating their osmotic potential, reducing the effect of toxic ions, and stimulating the antioxidant enzyme system [15]. Its application has shown great potential to mitigate the effects of salt stress on various agricultural crops and other plants [16]. However, its effects on nasturtium remain unknown.

Considering the current scenario of climate change and the reduction in water available for agriculture, the need to use saline waters in irrigation is increasing. Thus, we hypothesize that the application of proline will reduce the effects of salt stress on nasturtium, while also improving the phytochemical quality of its flowers. In this context, this study aimed to evaluate the effect of proline application on the growth, gas exchange, and phytochemical quality of nasturtium.

## 2. Results

This study tested the hypothesis that proline application mitigates the deleterious effects of salinity. To evaluate this, growth parameters, gas exchange, chlorophyll fluorescence, sugar content, and antioxidant levels were analyzed in nasturtium plants grown under protected environmental conditions.

The different electrical conductivities of water had a significant effect (*p* ≤ 0.01) on stem diameter, number of leaves, and root length. Additionally, proline application influenced plant height, stem diameter, and the number of nasturtium leaves.

Proline application increased plant height, stem diameter, and number of leaves in nasturtium (Figure 1A–C). The highest value of plant height (24.0 cm) at 60 DAS was achieved with a proline dose of 15.0 mM (Figure 1A). In addition, plants that received 15.0 mM of proline showed a 4.8 mm increase in stem diameter and a 12.0 leaf increase in the number of leaves compared to plants that received no proline application (Figure 1B,C).

The presence of salinity in irrigation water had a negative impact on the growth of nasturtium seedlings. Linear reductions in stem diameter, number of leaves, and root length were observed as the amount of salts added to the irrigation water increased, resulting in decreases of 22.41, 32.57, and 43.65% when comparing the lowest and highest salinity levels (Figure 1D–F).

The gas exchange of nasturtium plants was significantly influenced by the interaction between the use of salinized water and the application of proline (Figure 2A–C). Plants irrigated with saline waters of 0.67, 2.46, and 1.34 dS m^−1^ and under proline concentrations of 15.0, 9.31, and 15 mM showed, respectively, maximum values of net photosynthesis (6.35 μmol CO_2_ m^−2^ s^−1^), transpiration (2.24 mmol H_2_O m^−2^ s^−1^), and stomatal conductance (0.075 mol H_2_O m^−2^ s^−1^) (Figure 2A–C). On the other hand, the combination of higher concentrations of proline and higher electrical conductivities reduced gas exchange.

Regarding the instantaneous carboxylation efficiency, there was a significant effect of proline doses (Figure 3A). Plants irrigated with water of 0.00 dS m^−1^ and under proline dose of 15.0 mM showed higher instantaneous carboxylation efficiency (0.026), while water use efficiency (WUE) was higher in plants free of proline and salt stress (Figure 3B).

It can also be observed that the addition of salts to the irrigation water promoted an increase in the values of iWUE, especially at the lowest doses of proline (0.0 mM), with a maximum iWUE of 89.32 obtained with water of 4.25 dS m^−1^ (Figure 3C).

A proline dose of 8.27 mM resulted in a maximum estimated Fm value of 0.58 when plants were irrigated with water of 2.68 dS m^−1^ (Figure 3D). However, higher salt concentrations and proline doses reduced Fm. Despite this, Fv/Fm was not affected by salinity or proline application.

Proline application reduced the detrimental effects of salt stress and promoted increments in fresh leaf mass, fresh stem mass, and fresh root mass in nasturtium plants (Figure 4A–C). The highest values of fresh leaf mass (8.97 g), fresh stem mass (23.06 g), and fresh root mass (0.925 g) were achieved with a proline dose of 15.0 mM under irrigation with water of 0.44, 1.12, and 2.68 dS m^−1^, respectively (Figure 4A–C).

The highest accumulation of dry leaf mass was observed in plants under no application of proline and irrigated with water of 6.5 dS m^−1^ (Figure 4D). On the other hand, dry stem mass was higher in plants receiving no application of saline water with a proline concentration of 15.0 mM (Figure 4E), whereas the accumulation of dry root mass was greater with the use of water of 2.68 dS m^−1^ and application of 15.0 mM of proline (Figure 4F).

Proline application resulted in a significant increase in the number and mass of nasturtium flowers. The highest number of flowers (10.60) and greatest flower mass (0.72 g) were achieved with a proline dose of 15.0 mM (Figure 5A,B).

The contents of these components increased with the addition of salts to the irrigation water and an increase in proline doses. For example, the maximum amount of reducing sugars was 3.07 mg^−1^ when 15.0 mM of proline was used in conjunction with an electrical conductivity of 6.27 dS m^−1^ (Figure 6A). Similarly, the highest soluble sugar content (9.76 mg^−1^) was observed in plants treated with water of 0.5 dS m^−1^ and 7.75 mM of proline, respectively (Figure 6B).

Proline application at a dose of 8.79 mM increased the content of total phenolic compounds, which reached 891.06 mg^−1^ in nasturtium flowers without irrigation with saline water (Figure 6C). On the other hand, the increase in the electrical conductivity of irrigation water resulted in reductions in the values of reducing and soluble sugars and in the values of phenolic compounds.

An average vitamin C content of 59.27 mg 100 g^−1^ of FM was observed at a proline dose of 15.0 mM (Figure 6D), together with a conductivity of 4.75 dS m^−1^. These results show the positive influence of proline and electrical conductivity on the synthesis and accumulation of vitamin C in plants.

## 3. Discussion

Osmoprotectants, such as proline, may improve salt stress tolerance in plants [17]. However, there is little information available on the mechanism of action of the application of this amino acid in nasturtium cultivated under salinity conditions. In this study, the application of proline reduced the harmful effects of salt stress on nasturtium physiology. Proline application resulted in increments in height, stem diameter, and root length (Figure 1A–C), in addition to reducing the harmful effects of salinity on net photosynthesis, transpiration, and stomatal conductance (Figure 2). This is because proline acts as an antioxidant and signal regulator, performing multiple functions that are vital for the adaptability of plants to stress [18].

Several studies have shown reductions in CO_2_ absorption, stomatal conductance, transpiration, and internal CO_2_ concentration in vegetables such as tomato [19], basil [9,10,11,12,13,14,15,16,17,18,19,20], and melon [21,22] in response to abiotic stresses. In this context, the effects of salinity on gas exchange varied according to the electrical conductivity of the irrigation water, resulting in an increase in net photosynthesis, transpiration, and stomatal conductance under an electrical conductivity of up to 3.00 dS m^−1^, and a decrease at higher conductivities (6.5 dS m^−1^). In addition, gas exchange was also increased by the addition of proline at levels of up to 15.0 mM.

The higher stomatal conductance (gs) observed in nasturtium plants when exposed to salt stress and proline application can be attributed to this osmoprotectant, which quickly penetrates cells and contributes to the maintenance of cellular turgidity, thus preserving the structure and function of the photosynthetic system under stress [14]. Additionally, this response may be associated with an increase in the relative water content in the leaves induced by the application of proline; however, our results did not show an effect of the treatments on RWC.

Net photosynthesis (A) was higher in plants that grew with the application of proline and in irrigation water of different electrical conductivities (Figure 2A). This may have been due to the higher gs in this treatment, in addition to the fact that salinity did not affect the transpiration rate. On the other hand, the combination of higher doses of proline and higher electrical conductivities of water reduced gas exchange. The fact that the highest A was observed in plants grown under saline conditions is related to stomatal control, which is considered the main physiological factor to optimize water use during water deficits, avoiding excessive water losses under conditions of prolonged drought [23] and leading to a consequent decrease in the transport of assimilates [24].

An increase in Fm was observed in plants subjected to moderate salinity (2.68 dS m^−1^). This suggests that the photosynthetic apparatus was not damaged and that the energy available for the photochemical reactions was not underutilized. In addition, the increase in Fm directly influences Fv, which represents the potentially active energy in photosystem II, resulting in an increase in capacity for transferring excitation energy [25,26]. However, as salinity increased to 6.5 dS m^−1^, there was a reduction in the maximum fluorescence values, indicating damage to the photosynthetic apparatus of plants exposed to higher levels of salinity. In addition, the use of proline at concentrations of up to 10.86 mM resulted in an increase in Fm under conditions of moderate salinity.

In this study, the use of proline did not attenuate the adverse effects of moderate salt stress on biomass production in nasturtium, only promoting biomass accumulation in the absence of stress. This effect can be attributed to the role of amino acids in the promotion of cell division, improvements in photosynthetic efficiency, and the accumulation of photoassimilates [27]. In addition, the results showed that proline increased in the fresh mass of leaves, stems, and roots (Figure 4), both in nasturtium plants subjected to moderate salt stress and in plants not subjected to such conditions. The highest fresh mass value (8.97 g) was achieved with the proline concentration of 15.0 mM (Figure 4A).

The irrigation of plants with saline water often results in reduced growth and dry mass accumulation [28,29]. However, the application of proline at a concentration of 15.0 mM promoted an increase in dry root mass of up to 2.68 dS m^−1^ (Figure 4F). This effect may be associated with the high concentration of nitrogen present in proline [30], which, as a component of several compounds, probably contributed to the growth and metabolism of nasturtium plants, as well as increasing their tolerance to salt stress [31,32]. Similar results were observed in nasturtium plants subjected to a water deficit, where low-molecular-weight nitrogen, supplemented with spermine, attenuated the adverse effects of salinity on plant growth, photosynthetic rate, and nutrient uptake [10,11,12,13,14,15,16,17,18,19,20,21,22,23,24,25,26,27,28,29,30,31,32,33].

The influence of proline in attenuating the detrimental effects of salinity on reducing and soluble sugars in nasturtium (Figure 6A,B) is associated with this amino acid’s ability to play a crucial role in the osmoregulation mechanism. This mechanism involves the close interaction of non-toxic elements with various cellular components, contributing to osmotic regulation and turgor maintenance, and thus ensuring sufficient water content in cells [34]. The accumulation of compatible solutes represents an essential strategy for osmoregulation and osmotic adjustment in the face of salt stress [35]. In addition, the impact of proline on sugar levels may be related to the preservation of net photosynthesis in plants subjected to salt stress, a primordial step in the accumulation of sugars in plants.

The addition of proline resulted in an increase in the total content of phenolic compounds, probably due to its ability to (i) activate enzymes in the biosynthesis of phenolic compounds, (ii) modulate defense-related gene expression, (iii) reinforce the response to salt stress, and (iv) amplify the hormonal signals of ABA and ethylene [36]. The regulation of these osmolytes may represent an effective mechanism to increase plant tolerance to salt stress caused by the accumulation of salts in the root zone, thus maintaining adequate cellular water homeostasis for a healthy metabolism [37]. The increase in salt stress resulted in a decrease in the content of phenolic compounds, since these compounds are a non-enzymatic pathway for the elimination of ROS and for the regulation of osmotic potential [38].

Proline application and saline water irrigation increased the accumulation of vitamin C in plants, which is an expected effect under stress conditions, since ascorbic acid is involved in the protection of lipids and proteins against salt-induced oxidative damage, neutralizing ROS, regulating stomatal movement and transpiration, protecting structures of photosynthesis and photosynthetic pigments, regulating ionic and osmotic balance within plant organs, and activating various antioxidant enzymes and phytohormones [39]. These values are close to those found in fruits considered important sources of this vitamin, such as orange (*Citrus cinensis*) (62.5 mg for every 100 g^−1^ of FM) and guava (*Psidium guajava*) (85.9 mg for every 100 g^−1^ of FM) [40]. High levels of vitamin C are also reported for some specific flowers, such as pansy (*Viola* × *wittrockiana*) (256 mg for every 100 g^−1^ of FM), chives (*Allium schoenoprasum*) (108 mg for every 100 g^−1^ of FM), and China pink (89.78 mg for every 100 g^−1^ of FM) [41,42,43].

## 4. Materials and Methods

### 4.1. Experimental Location

The experiment was carried out in a greenhouse located in the experimental area of the Biotechnology and Plant Breeding Sector of the Biosciences Department of the Center for Agrarian Sciences, Federal University of Paraíba, Areia, Paraíba, Brazil, whose geographic coordinates are 6°57′48″ S and 35°41′30″ W, with an altitude of 618 m. According to Köppen’s classification, the local climate is ‘As’ type, with dry and hot drought periods and rainfall in winter [44].

During the experiment (August to October 2023), the minimum and maximum daily temperatures and the relative humidity of the air were recorded using a digital thermo-hygrometer (AKSO^®^ AK28new, Sao Leopoldo, RG, Brazil) (Figure 7). The photoperiod during the experiment corresponded to 12 h/12 h of light/dark.

### 4.2. Experimental Design and Conduction

The experimental design was completely randomized in a 5 (electrical conductivities—ECw: 0.0, 1.50, 3.00, 4.5, 6.5 dS m^−1^) × 4 (proline concentrations—Pro: 0, 5, 10, 15 mM) factorial scheme, with six replicates. The electrical conductivities of water were chosen based on [45], while proline concentrations were adapted according to [46].

The plants were grown in 5.0 dm^3^ polyethylene pots. Nasturtium (*Tropaeolum majus*, cv. assorted dwarf—Isla^®^, Porto Alegre, RS, Brazil) seeds were planted, two per pot, at a depth of approximately 2 cm. Thinning was performed at 10 days after sowing (DAS), keeping the most vigorous plant in each pot. The substrate used was Mecplant^®^ (Telêmaco Borba, PR, Brazil), which is composed of 60% pine bark, 15% fine vermiculite, 15% superfine vermiculite, and 10% humus. A chemical characterization of the substrate used is presented in Table 1.

Prior to irrigation with saline waters, standardization was carried out for height, diameter at collar height, and two pairs of leaves to reduce the heterogeneity in the experiment during plant establishment production and to ensure fruit quality. The saline waters were prepared by adding sodium chloride (NaCl) to the water until they reached the established electrical conductivities, which was checked using a portable microprocessor-based conductivity meter (CD-860 model, Instrutherm^®^, Sao Paulo, SP, Brazil) to determine the values. From 15 DAS onwards, irrigation with saline water was carried out manually according to the water needs of the plants, which were established via the drainage lysimetry method [47].

Proline solutions were prepared in distilled water, with the addition of Tween 80 (0.05%) as a surfactant to increase absorption by plants. In plants that were not treated with proline (0 mM), the surfactant Tween 80 was also applied (0.05%). Proline was foliar-applied every 10 days after irrigation with saline water (25, 35, 45 and 55 DAS), according to the established doses. The applications were carried out with a manual sprayer, distributing the product on the adaxial and abaxial surfaces of the leaves. In the first application, all plants were sprayed with 2.5 mL, in the second application, they were sprayed with 4.0 mL, in the third, with 5.5 mL, and in the fourth application, with 7.5 mL of each solution, totaling 19.5 mL being applied per plant during the experimental period.

Analyses of growth and physiological aspects were carried out 60 days after the beginning of irrigation with saline water.

### 4.3. Plant Growth

Plant height was evaluated by measuring the distance from the collar to the last leaf incision of the plant with a graduated ruler, and the values were expressed in cm. Stem diameter was measured with a digital caliper and the values were expressed in mm. Number of leaves was determined by counting all leaves per plant. Root length was obtained by measuring the distance between the stem surface and the tip of the root using a graduated ruler, with values expressed in cm.

### 4.4. Gas Exchange

Gas exchange was determined with an infrared gas analyzer (IRGA, LCpro-SD Portable Photosynthesis System, ADC BioScientific, Hoddesdon, UK). The measurements were carried out from 8 to 10 am, using artificial light fixed at 1000 μmol photons m^−2^ s^−1^, CO_2_ concentration and ambient temperature. Stomatal conductance (gs—mol H_2_O m^−2^ s^−1^), net photosynthesis (A—μmol CO_2_ m^−2^ s^−1^), transpiration (E—mmol H_2_O m^−2^ s^−1^), water use efficiency (WUE = A/E), intrinsic water use efficiency (iWUE = A/gs), internal carbon concentration (Ci—μmol CO_2_ mol air^−1^), and instantaneous carboxylation efficiency (iCE = A/Ci) were evaluated in leaves located in the middle third.

Initial (F_0_), maximum (Fm) and variable (Fv) fluorescence and the quantum yield of photosystem II (Fv/Fm) were measured between 8 and 11 am, using a modulated fluorometer (OptiSciences Inc.—OS-30p model, Hudson, NY, USA), in one leaf in the middle third per plant, after being pre-adapted to the dark for 30 min. Chlorophyll *a*, chlorophyll *b*, and total chlorophyll indices were measured on the third leaf from the apex of each plant with a digital chlorophyll meter (ClorofiLOG^®^, CFL 1030 model, Porto Alegre, RS, Brazil).

At the end of the experiment, at 60 DAS, the plants were harvested and partitioned into the root, leaves, and stem. Their fresh mass was determined using a precision analytical scale (0.001 g), and then the different parts were packed in Kraft paper bags and dried in the oven with forced air circulation at a temperature of 65 °C until a constant weight was reached. Plant dry mass was measured with a precision analytical scale (0.001 g).

### 4.5. Relative Water Content

Relative water content (RWC) was calculated according to the methodology described by [46], determined from 10 leaf disks of 0.5 cm diameter each, collected from fully expanded leaves. The disks were weighed on a precision scale immediately after collection to obtain fresh mass (FM). Then, the disks were transferred to Petri dishes, where they remained submerged in water for six hours. After this period, the disks were weighed again to determine the turgid mass (TM) and were subsequently dried at 70 °C for 72 h to obtain the dry mass (DM). RWC was calculated using the formula:RWC (%) = (FM − DM)/(TM − DM) × 100
where RWC is relative water content, FM is the fresh mass of the disks, TM is the turgid mass of the disks, and DM is the dry mass of the disks.

### 4.6. Physicochemical Analyses

At the Laboratory of Post-Harvest Physiology—CCHSA/UFPB, physicochemical analyses of the flowers were carried out, looking at their pH, acidity, soluble solids (expressed in °Brix), vitamin C, total phenolic compounds, and reducing and non-reducing sugars [48]. Titratable acidity (% of citric acid) was determined by diluting 10 g of flower pulp in 50 mL of distilled water. The solution was then titrated with 0.1 N NaOH solution until it became a light pinkish color. The results were expressed as a percentage of citric acid according to the method of the Adolfo Lutz Institute [49].

pH determination was performed by weighing 10 g of a macerated flower sample and adding 50 mL of distilled water. The solution was then measured with a digital potentiometer (HANNA, Singapore), according to the method of the Association of Official Analytical Chemists [50]. Soluble solids content (%) was evaluated after the maceration of the sample in a multiprocessor. A drop of the crushed pulp was placed in a refractometer and the results were expressed in °Brix, as in Reference [49]. The amount of total vitamin C (100 g^−1^ mg of flowers) was determined via spectrophotometry, with a reading at 520 nm, according to the method of [51].

The amount of reducing sugars was determined via the 3,5-dinitrosalicylic acid (DNS) method, as described by [52]. The extract was prepared by diluting 1 g of pulp in 50 mL of distilled water. An aliquot of 0.8 mL of the extract was mixed with 0.7 mL of water and 1.0 mL of the dinitrosalicylic acid solution to obtain the samples. The samples were shaken and kept in a water bath at 100 °C for 5 min. The standard curve was prepared with glucose, and sample readings were taken in a spectrophotometer at 450 nm. The amount of non-reducing sugars was calculated by the difference between the total sugars and the reducing sugars. The total sugars (g 100 g^−1^ of flowers) were determined using the anthrone method, according to the methodology described by [53]. The extract was obtained by diluting 0.5 g of flowers in 100 mL of distilled water. The samples were prepared in an ice bath, with the addition of 200 μL of the extract, 800 μL of distilled water, and 2.0 mL of the 0.2% anthrone solution. Then, the samples were shaken and kept in a water bath at 100 °C for 8 min. The flower samples were examined with a spectrophotometer at 620 nm, using glucose as a reference, to obtain the standard curve.

Total vitamin C (100 g^−1^ mg of nasturtium flowers) was determined according to [54]. Total phenolics were determined using the spectrophotometric method of [55]. Absorbance readings were performed at a wavelength of 760 nm, and the results were expressed in milligrams of gallic acid equivalent (GAE) per gram of sample.

The extraction and quantification of chlorophylls was performed according to the methodology proposed by [56]. Five leaf disks (1 cm in diameter each) were collected and incubated in 5 mL of dimethyl sulfoxide (DMSO). The samples were kept in the dark for 48 h at room temperature. After incubation, the absorbance of the samples (480, 649, and 665 nm) was determined using a glass cuvette with an optical path of 10 mm using a spectrophotometer. The wavelengths and equations used to calculate the concentrations of chlorophyll *a*, chlorophyll *b*, and carotenoids are based on the method described by [57].

### 4.7. Statistical Analyses

The data obtained were subjected to analysis of variance (*p* ≤ 0.05) and, when significant differences were obtained, a regression analysis was performed. The data were processed using the R language with RStudio interface (version 4.2.1) [58]. The graphs were created in SigmaPlot^®^ 12.5 software (Systat Software, San Jose, CA, USA).

## 5. Conclusions

The application of 15.0 mM proline positively influenced nasturtium by enhancing its growth parameters, including the number of leaves, the stem diameter, the root length, and the mass and number of flowers. When combined with moderate salinity (3.0 dS m^−1^), this concentration of proline also stimulated gas exchange, leading to increased net photosynthesis, transpiration, and stomatal conductance. Additionally, proline elevated the levels of total phenolic compounds, reduced sugars, and soluble sugar content, while also boosting vitamin C activity, particularly in plants exposed to severe salt stress (4.75 dS m^−1^). These results are based on the applied concentrations of proline and were obtained under controlled greenhouse conditions, which may vary in the field.

## Figures and Tables

**Figure 1 plants-14-00301-f001:**
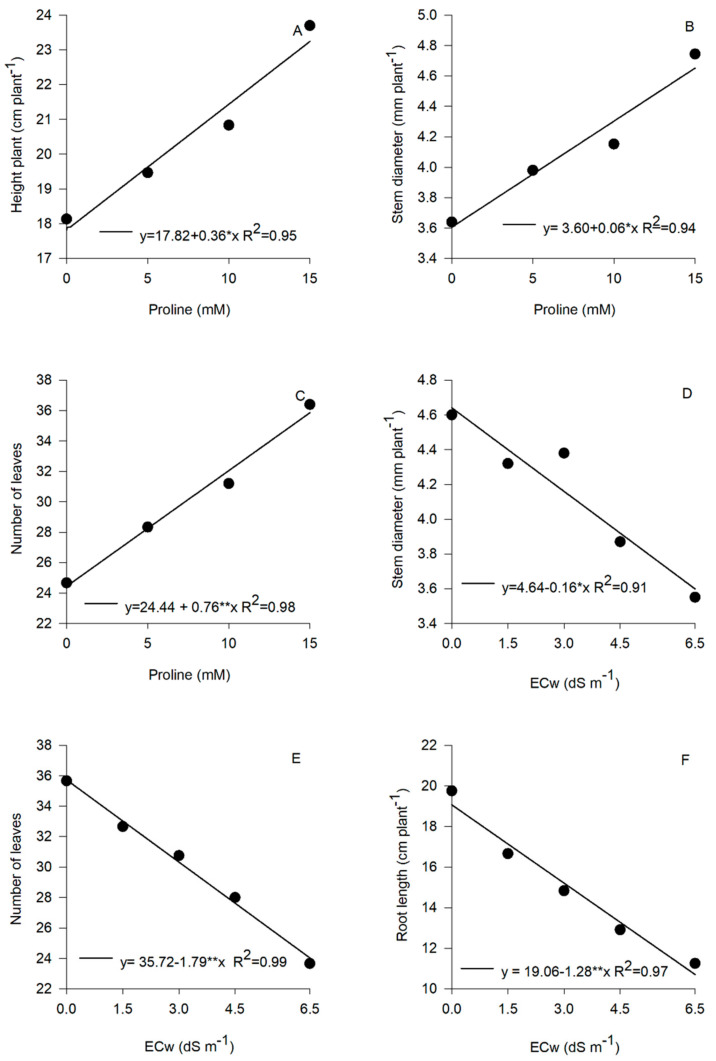
Plant height (**A**), stem diameter (**B**), and number of leaves (**C**) under proline application. Stem diameter (**D**), number of leaves (**E**), and root length (**F**) under various electrical conductivities of irrigation water (ECw). Plants were measured 60 days after sowing. Asterisks indicate significant differences determined via the F-test at 5% (*) or 1% (**) probability levels. (*n* = 6).

**Figure 2 plants-14-00301-f002:**
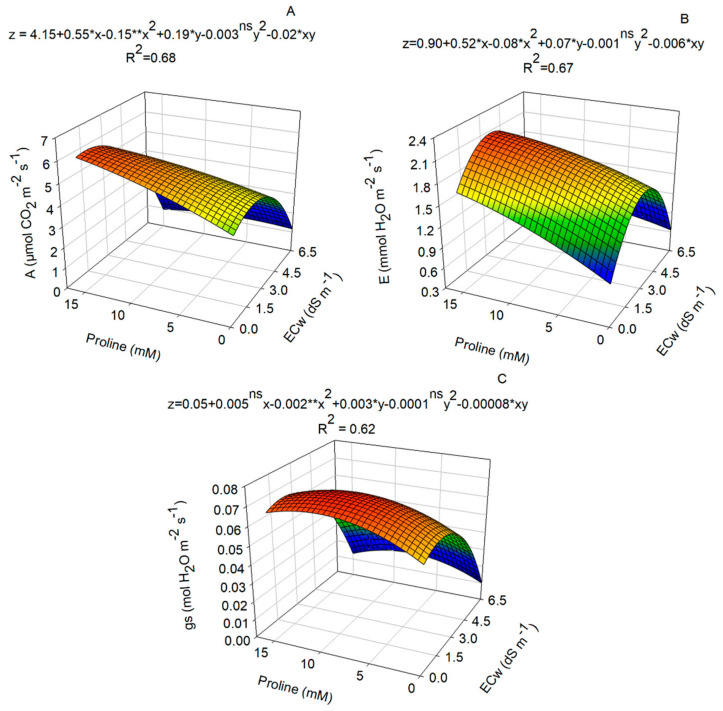
Gas exchange of nasturtium plants, cv. Anã sortida, subjected to different electrical conductivities of irrigation water (ECw) and proline concentrations. (**A**) Net photosynthesis (A), (**B**) transpiration (E), and (**C**) stomatal conductance (gs). Plants were measured 60 days after sowing. Asterisks indicate significant differences determined via the F-test at 5% (*) or 1% (**) probability levels. ^ns^: Not significant (*n* = 6).

**Figure 3 plants-14-00301-f003:**
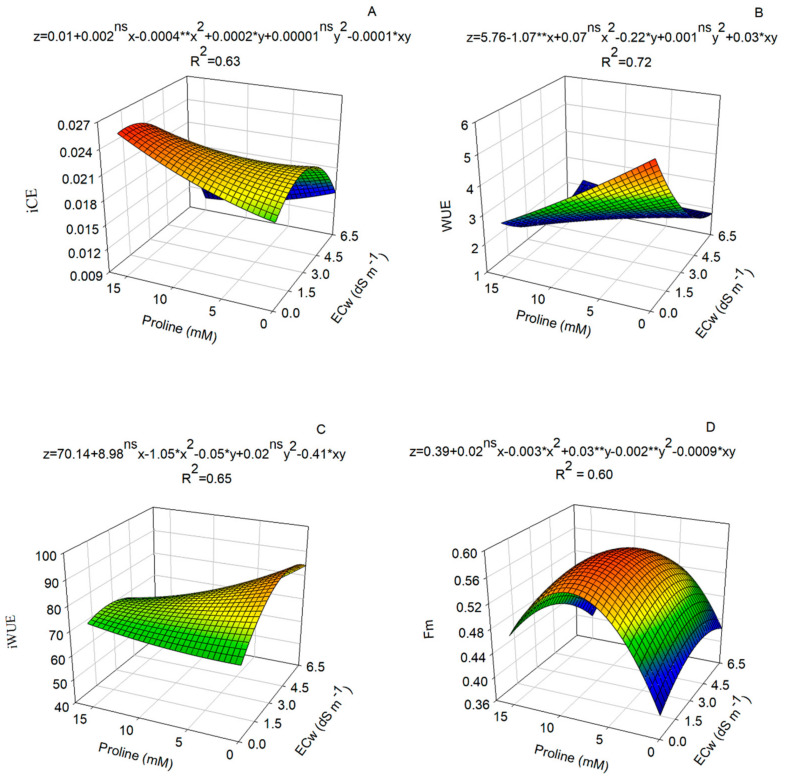
Gas exchange of nasturtium plants, cv. Anã sortida, subjected to different electrical conductivities of irrigation water (ECw) and proline concentrations. (**A**) Instantaneous carboxylation efficiency (iCE); (**B**) water use efficiency (WUE); (**C**) intrinsic water use efficiency (iWUE), and (**D**) maximum fluorescence (Fm). Plants were measured 60 days after sowing. Asterisks indicate significant differences determined via the F test at 0.05 (*) or 0.01 (**) probability levels. ^ns^: Not significant (*n* = 6).

**Figure 4 plants-14-00301-f004:**
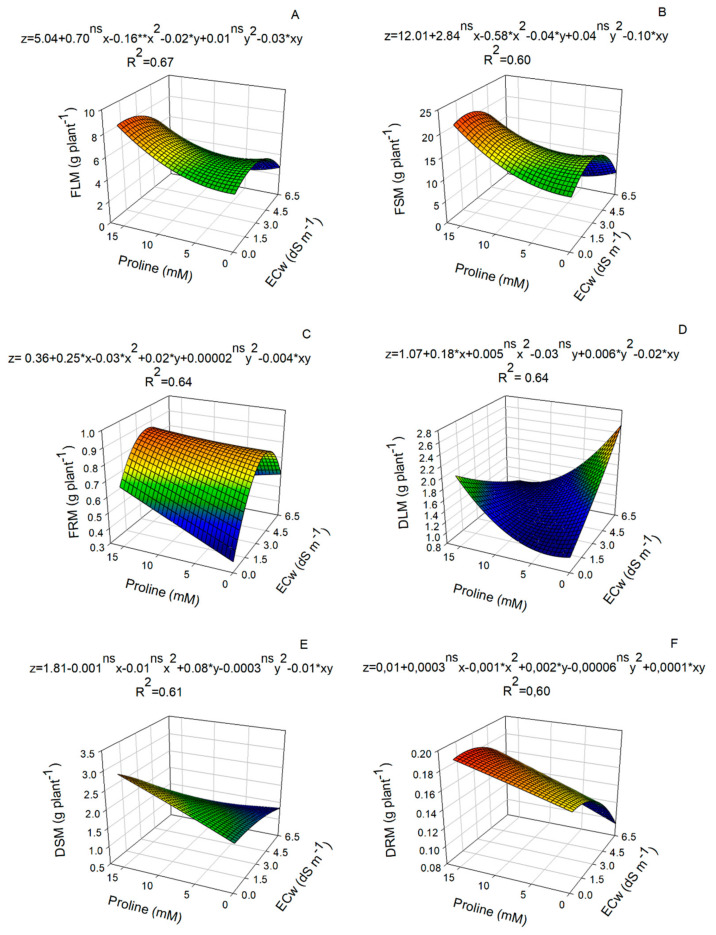
Biomass production in nasturtium plants, cv. Anã sortida, in response to different electrical conductivities of irrigation water (ECw) and proline doses. Fresh leaf mass (**A**); fresh stem mass (**B**); fresh root mass (**C**); dry leaf mass (**D**); dry stem mass (**E**); and dry root mass (**F**). Plants were measured 60 days after sowing. ^ns^, *, ** not significant, significant at *p* ≤ 0.05 and *p* ≤ 0.01 as determined via the F test, respectively (*n* = 6).

**Figure 5 plants-14-00301-f005:**
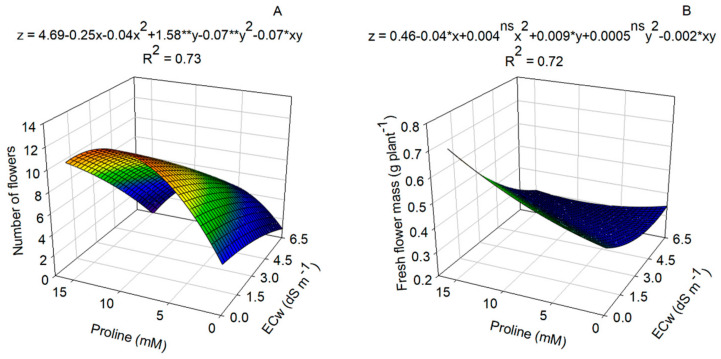
Number of flowers (**A**) and flower mass (**B**) in nasturtium plants subjected to irrigation with salinized water and proline doses at 60 days after sowing. ^ns^, *, ** not significant, significant at *p* ≤ 0.05 and *p* ≤ 0.01 as determined via the F test, respectively (*n* = 6).

**Figure 6 plants-14-00301-f006:**
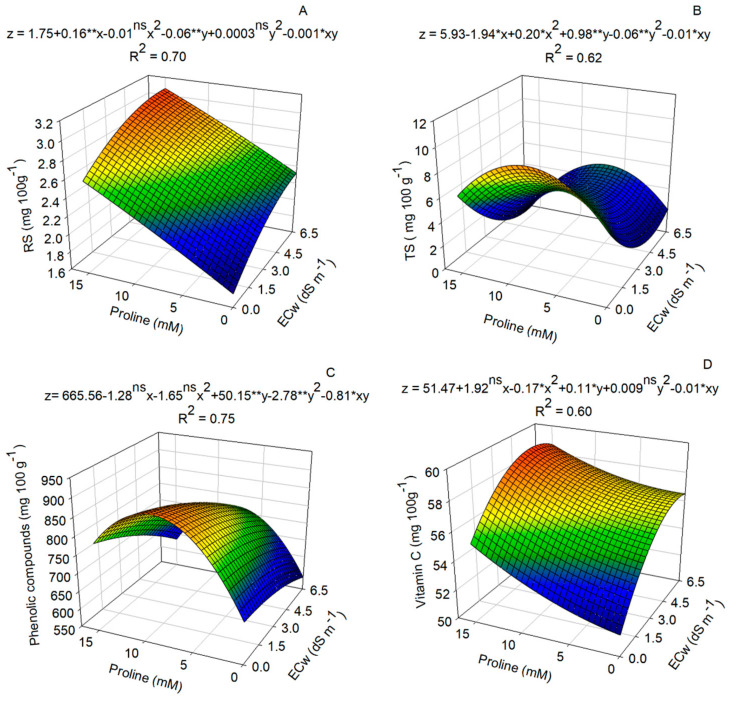
Reducing sugars (**A**), total soluble sugars (**B**), phenolic compounds (**C**), and vitamin C (**D**) in nasturtium plants subjected to irrigation with salinized water and proline doses at 60 days after sowing. ^ns^, *, ** not significant, significant at *p* ≤ 0.05 and *p* ≤ 0.01 as determined via the F test, respectively (*n* = 6).

**Figure 7 plants-14-00301-f007:**
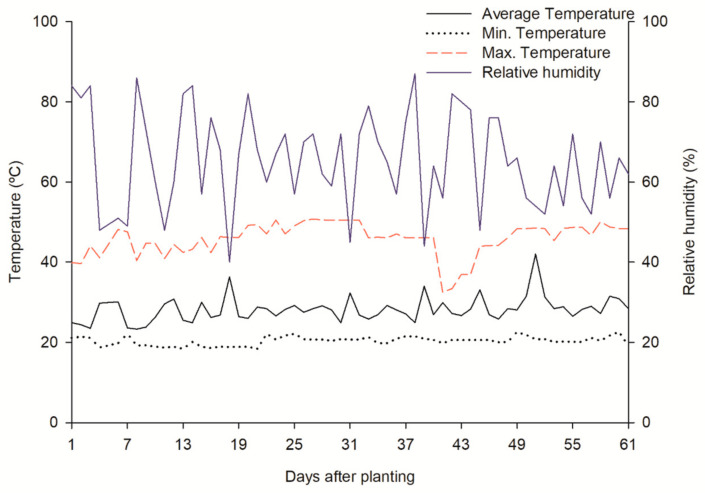
Temperature and relative humidity data during the experimental period.

**Table 1 plants-14-00301-t001:** Chemical characterization of the substrate used in the experiment.

pH_(H2O,1:2.5)_	P	Na	H + Al	Al	Ca	Mg	K	C
	mg kg^−1^	mg kg^−1^	---------cmol_c_ kg^−1^---------				mg kg^−1^	g kg^−1^
5.00	233.27	14.69	20.62	0	8.9	8.9	97.59	35.25

P and K—Mehlich extraction; C—Carbon; Al, Ca, and Mg—KCL; H + Al—Calcium acetate.

## Data Availability

All data generated and/or analyzed during the present study are available in the manuscript and the corresponding authors have no objection to making the data and materials available upon reasonable request.

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
