# Peer review of "Growth, Gas Exchange, and Phytochemical Quality of Nasturtium (Tropaeolum majus L.) Subjected to Proline Concentrations and Salinity"

_plants, 2025, doi:10.3390/plants14030301_

Round 1
Reviewer 1 Report
Comments and Suggestions for Authors
Review on „Growth, gas exchange, and phytochemical quality of nasturtium (Tropaeolum majus L.) subjected to proline concentrations and salinity”
Proline, a small amino acid with a unique cyclic structure, plays a multifaceted role in plant physiology. Beyond its primary function as a proteinogenic amino acid, proline acts as a crucial molecule in stress responses, metabolic regulation, and development. It is particularly notable for its role as an osmoprotectant and signaling molecule under abiotic stresses such as drought, salinity, and temperature extremes. This study examined the effect of different electrical conductivities and proline concentrations on nasturtium. My main concern about the manuscript is that no explanation of why these specific electrical conductivities and proline concentrations were applied in the experiment. Please explain it in the Introduction section or use a reference that used the same treatments in the Materials and Methods section. My second concern is that the authors applied different proline concentrations but did not measure proline in plant tissues. Why is that? How do they know how much proline was uptaken/absorbed by the plants?
The manuscript's research area aligns with the Plants-MDPI journal's scope. The manuscript does, however, require a few small edits.
Please check the Oxford comma throughout the manuscript (e.g. lines 77, 78, 83, 85, etc.).
The titles of each table and figure should contain the number of repetitions of the measured parameter. Please add the information.
Abstract:
Line 14. Crop usually refers to a cultivated plant that is grown as food, especially a grain, fruit, or vegetable. Nasturtium’s leaf and flower are consumed.
Keywords:
Please arrange the keywords in alphabetical order.
Results:
Please check the numbering of the figures. Figure 2 must be Figure 1. Figure 1 must be Figure 7, etc.
Line 78: “The highest value of plant height (24.0 cm) at 60 DAS…” Each figure and table must be understandable by itself without reading the whole manuscript. Please add the information the the title of Figure 2 that the data presented at 60 days after sowing (60 DAS). Please check and correct the other figures as well.
Figure 2. In the case of proline treatment, the authors measured the plant height, stem diameter, and the number of leaves. While in the case of „salinity” treatment, stem diameter, the number of leaves, and root length were measured. Why weren't the same parameters measured for all treatments? The title of Figure 2 explains that „Stem diameter (d), number of leaves (e) and root length (f) under electrical conductivities of irrigation 118 water (ECw) and proline concentrations.” However, the titles of x-axis (Figures 2D, 2E, and 2F) did not show the proline treatments. It is confusing please correct it.
Materials and Methods:
Lines 291-299: Please add information about the length of day and night in the greenhouse during the experiment.
Line 305. How were the applied electrical conductivities and proline concentrations selected? Please add references and/or explain why these specific treatments were selected.
Line 307: „The seedlings were produced in 5.0-dm3 polyethylene pots.” I think this sentence is not correct. Please revise it.
Conclusion:
The authors did not measure the proline content in plant tissue so the results and conclusions were based on the applied proline concentrations. Please include this explanation in the conclusion section. In addition, please also indicate that these research findings and conclusions are based on a greenhouse experiment under controlled environmental conditions. These data can be varied under field conditions.
Author Response
REVIEWER #1:
Review on “Growth, gas exchange, and phytochemical quality of nasturtium (Tropaeolum majus L.) subjected to proline concentrations and salinity”
Proline, a small amino acid with a unique cyclic structure, plays a multifaceted role in plant physiology. Beyond its primary function as a proteinogenic amino acid, proline acts as a crucial molecule in stress responses, metabolic regulation, and development. It is particularly notable for its role as an osmoprotectant and signaling molecule under abiotic stresses such as drought, salinity, and temperature extremes. This study examined the effect of different electrical conductivities and proline concentrations on nasturtium. My main concern about the manuscript is that no explanation of why these specific electrical conductivities and proline concentrations were applied in the experiment. Please explain it in the Introduction section or use a reference that used the same treatments in the Materials and Methods section.
Response: We greatly appreciate the thorough revision of our manuscript. The references used to choose the proline concentrations and electrical conductivities were added to the Material and Methods section, as suggested (lines 318-319).
My second concern is that the authors applied different proline concentrations but did not measure proline in plant tissues. Why is that? How do they know how much proline was uptaken/absorbed by the plants?
Response: We acknowledge the importance of measuring endogenous proline levels; however, our study lacked the necessary tools to perform such assessments. Nonetheless, we reaffirm that the novelty of applying this approach to this species provides an innovative and practical contribution to the field.
The manuscript's research area aligns with the Plants-MDPI journal's scope. The manuscript does, however, require a few small edits.
Please check the Oxford comma throughout the manuscript (e.g. lines 77, 78, 83, 85, etc.).
Response: We thank the reviewer for the valuable comments. We have now revised the entire manuscript, double-checking the use of Oxford coma.
The titles of each table and figure should contain the number of repetitions of the measured parameter. Please add the information.
Response: Thanks for your comment. We have added the number of repetitions in the figure captions, as suggested.
Abstract:
Line 14. Crop usually refers to a cultivated plant that is grown as food, especially a grain, fruit, or vegetable. Nasturtium’s leaf and flower are consumed.
Response: We thank the reviewers for bringing this to our attention, the term "crop" has been removed in this context throughout the text.
Keywords:
Please arrange the keywords in alphabetical order.
Response: Rearranged, as suggested.
Results:
Please check the numbering of the figures. Figure 2 must be Figure 1. Figure 1 must be Figure 7, etc.
Response: Corrected, as suggested.
Line 78: “The highest value of plant height (24.0 cm) at 60 DAS…” Each figure and table must be understandable by itself without reading the whole manuscript. Please add the information the title of Figure 2 that the data presented at 60 days after sowing (60 DAS). Please check and correct the other figures as well.
Response: Corrected, as suggested.
Figure 2. In the case of proline treatment, the authors measured the plant height, stem diameter, and the number of leaves. While in the case of „salinity” treatment, stem diameter, the number of leaves, and root length were measured. Why weren't the same parameters measured for all treatments? The title of Figure 2 explains that „Stem diameter (d), number of leaves (e) and root length (f) under electrical conductivities of irrigation 118 water (ECw) and proline concentrations.” However, the titles of x-axis (Figures 2D, 2E, and 2F) did not show the proline treatments. It is confusing please correct it.
Response: Actually, as the experimental design was in a factorial scheme, the data presented are those whose factors were significant, so in variables where both factors were significant such as the number of leaves, both proline and salinity are shown and when only one of these factors is significant only this one is shown.
Materials and Methods:
Lines 291-299: Please add information about the length of day and night in the greenhouse during the experiment.
Response: Done as suggested. The information was added to lines 309-310.
Line 305. How were the applied electrical conductivities and proline concentrations selected? Please add references and/or explain why these specific treatments were selected.
Response: This information was added in the lines 317-318, as requested.
Line 307: The seedlings were produced in 5.0-dm3 polyethylene pots.” I think this sentence is not correct. Please revise it.
Response: Corrected, as suggested.
Conclusion:
The authors did not measure the proline content in plant tissue so the results and conclusions were based on the applied proline concentrations. Please include this explanation in the conclusion section. In addition, please also indicate that these research findings and conclusions are based on a greenhouse experiment under controlled environmental conditions. These data can be varied under field conditions.
Response: Added, as suggested.
Reviewer 2 Report
Comments and Suggestions for Authors
In this manuscript, the authors described the effect of proline application on the salt tolerance of nasturtium in a greenhouse setting. Although the application of proline to reduce salt stress in crops is not novel, the authors presented a thorough and well-supported research on this effect in nasturtium, an important crop species in Brazil. Overall, I support the publication of this research. I only have a few minor comments:
1. At the beginning of the results section, in a few sentences, please briefly describe how the experiment was performed, before describing the results.
2. In the methods, it’s worth noting whether or not the 0 mM proline control solution also contained 0.05% Tween 80 as the proline-containing treatment solutions.
3. Line spacing of line 257-287 needs adjustment to make it appear the same as the rest of the main text.
Author Response
REVIEWER #2:
In this manuscript, the authors described the effect of proline application on the salt tolerance of nasturtium in a greenhouse setting. Although the application of proline to reduce salt stress in crops is not novel, the authors presented a thorough and well-supported research on this effect in nasturtium, an important crop species in Brazil. Overall, I support the publication of this research. I only have a few minor comments:
- At the beginning of the results section, in a few sentences, please briefly describe how the experiment was performed, before describing the results.
Response: Added, as suggested (lines 77-80).
- In the methods, it’s worth noting whether or not the 0 mM proline control solution also contained 0.05% Tween 80 as the proline-containing treatment solutions.
Response: Thank you for bringing this to our attention, a proper explanation has been added on lines 339-340.
- Line spacing of line 257-287 needs adjustment to make it appear the same as the rest of the main text.
Response: Done as suggested.
Round 2
Reviewer 1 Report
Comments and Suggestions for Authors
Thank you for the corrections and responses.